# Combining IL-6 and SARS-CoV-2 RNAaemia-based risk stratification for fatal outcomes of COVID-19

Ryo Saji[1], Mototsugu Nishii[1]*, Kazuya Sakai[1], Kei Miyakawa[2], Yutaro Yamaoka[2,3], Tatsuma Ban[4], Takeru Abe[5], Yutaro Ohyama[1], Kento Nakajima[1], Taro Hiromi[1], Reo Matsumura[1], Naoya Suzuki[6], Hayato Taniguchi[5], Tsuyoshi Otsuka[6], Yasufumi Oi[1], Fumihiro Ogawa[1], Munehito Uchiyama[1], Kohei Takahashi[5], Masayuki Iwashita[5], Yayoi Kimura[7], Satoshi Fujii[8], Ryosuke Furuya[6], Tomohiko Tamura[4,7], Akihide Ryo[2,7], Ichiro Takeuchi[1,5]

1 Department of Emergency Medicine, Yokohama City University Graduate School of Medicine, Yokohama, Kanagawa, Japan, 2 Department of Microbiology, Yokohama City University Graduate School of Medicine, Yokohama, Kanagawa, Japan, 3 Life Science Laboratory, Technology and Development Division, Kanto Chemical Co., Inc., Isehara, Kanagawa, Japan, 4 Department of Immunology, Yokohama City University Graduate School of Medicine, Yokohama, Kanagawa, Japan, 5 Department of Advanced Critical Care and Emergency Center, Yokohama City University Medical Center, Yokohama, Kanagawa, Japan, 6 Department of Critical Care and Emergency Medicine, National Hospital Organization Yokohama Medical Center, Yokohama, Kanagawa, Japan, 7 Department of Advanced Medical Research Center, Yokohama City University, School of Medicine, Yokohama, Kanagawa, Japan, 8 Department of Molecular Pathology, Yokohama City University Graduate School of Medicine, Yokohama, Kanagawa, Japan

* s461211@yokohama-cu.ac.jp

## Abstract

### Background

The coronavirus disease 2019 (COVID-19) pandemic rapidly increases the use of mechanical ventilation (MV). Such cases further require extracorporeal membrane oxygenation (ECMO) and have a high mortality.

### Objective

We aimed to identify prognostic biomarkers pathophysiologically reflecting future deterioration of COVID-19.

### Methods

Clinical, laboratory, and outcome data were collected from 102 patients with moderate to severe COVID-19. Interleukin (IL)-6 level and severe acute respiratory syndrome coronavirus 2 (SARS-CoV-2) RNA copy number in plasma were assessed with ELISA kit and quantitative PCR.

### Results

Twelve patients died or required ECMO owing to acute respiratory distress syndrome despite the use of MV. Among various variables, a ratio of oxygen saturation to fraction of

**Data Availability Statement:** The data used in this paper were acquired from https://doi.org/10.6084/m9.figshare.15059814.

**Funding:** This study was funded by the Japan Agency for Medical Research and Development (JP19fk0108169). This funding organization did not play a role in the study design, data collection and analysis, decision to publish, or preparation of the manuscript and did not provide financial support in the form of authors' salaries. This funding only provided financial support in the form of research materials. YM, a contributor is employed by Kanto Chemical Co provided support in the form of salaries, but the Kanto Chemical Co did not have any role in this study, including study design, data collection and analysis, decision to publish, provision of financial support in the form of authors' salaries and research materials, and preparation of the manuscript. This does not alter our adherence to PLOS ONE policies on sharing data and materials.

**Competing interests:** The authors have declared that no competing interests exist.

inspired oxygen (SpO2/FiO2), IL-6, and SARS-CoV-2 RNA on admission before intubation were strongly predictive of fatal outcomes after the MV use. Moreover, among these variables, combining SpO2/FiO2, IL-6, and SARS-CoV-2 RNA showed the highest accuracy (area under the curve: 0.934). In patients with low SpO2/FiO2 (< 261), fatal event-rate after the MV use at the 30-day was significantly higher in patients with high IL-6 (> 49 pg/mL) and SARS-CoV-2 RNAaemia (> 1.5 copies/μL) compared to those with high IL-6 or RNAaemia or without high IL-6 and RNAaemia (88% vs. 22% or 8%, *log-rank test P* = 0.0097 or *P* < 0.0001, respectively).

## Conclusions

Combining SpO2/FiO2 with high IL-6 and SARS-CoV-2 RNAaemia which reflect hyperinflammation and viral overload allows accurately and before intubation identifying COVID-19 patients at high risk for ECMO use or in-hospital death despite the use of MV.

## Introduction

In humans, coronaviruses cause respiratory tract infections represented by severe acute respiratory syndrome (SARS) and the Middle East respiratory syndrome [1, 2]. In December 2019, pneumonia cases of an unknown etiology emerged in Wuhan, Hubei, China, with clinical presentations resembling viral pneumonia. Sequencing analysis of lower respiratory tract samples indicated the existence of a novel coronavirus, SARS coronavirus 2 (SARS-CoV-2) [3, 4]. This disease caused by SARS-CoV-2 was officially named "coronavirus disease 2019" (COVID-19). COVID-19 has spread rapidly worldwide and has been classified as a global pandemic [5]. Initial studies revealed that approximate 30% of affected patients were transferred to the intensive care unit due to complications including acute respiratory distress syndrome (ARDS), arrhythmia, and shock. The overall mortality of COVID-19 was 4–15% [6–8]. Disruption of medical care system by inadequate patient stratification raised the mortality rate of COVID-19. Particularly, restricting the population studies to patients who were provided mechanical ventilation (MV) for severe respiratory failure affects the evaluation of mortality negatively [6–8]. Such patients with critically ill COVID-19 may further require salvage therapy such as extracorporeal membrane oxygenation (ECMO) for severe ARDS leading to in-hospital death. Thus, risk stratification for fatal outcomes including ECMO use and in-hospital death despite the use of MV would keep the medical care system running efficiently, contributing to the improvement of mortality.

Synergistic cooperation between interferon (IFN)-gamma-induced T cell immune response and interleukin (IL)-6-induced macrophage activation in COVID-19 has been implicated in the development of ARDS [9, 10]. Consistently, high levels of circulating IL-6 have been associated with the need for MV and high mortality in COVID-19 patients [11, 12]. Alternatively, SARS-CoV-2 infection of immune cells mediates their activation and promotes the release of IL-6 and other inflammatory cytokines from them, which in turn results in ARDS [13]. Detectable SARS-CoV-2 RNA in peripheral blood (RNAaemia) has been observed in critically ill COVID-19 patients with drastic increase in circulating IL-6 levels [14]. High IL-6 and viral RNAaemia, which would reflect hyperinflammation and viral overload as its trigger, may contribute to accurate prediction of clinical outcomes based on pathophysiological understanding of future deterioration of COVID-19. To our knowledge, however, combining immunological

marker and viral RNA load for prediction of clinical outcomes is not common even in viral diseases and its clinical utility remains elusive.

Here, we aimed to evaluate the utility of combining IL-6 level and SARS-CoV-2 RNA load in early prediction of fatal outcomes in COVID-19 patients, including ECMO use and in-hospital death despite the use of MV.

## Materials and methods

### Study design

This study was a prospective observational study across three hospitals in Japan including Yokohama City University Hospital (YCUH), Yokohama City University Medical Center (YCUMC), and National Hospital Organization Yokohama Medical Center (NHOYMC). All consecutively admitted patients with the COVID-19 during February 2020 to July 2021 were enrolled in this study. The YCUH and YCUMC were mainly intended for severe to critical cases, while NHOYMC was for moderate to severe cases. Therefore, a wide range of cases from moderate to critical were included in the study population. Enrolled patients were observed until 30-day or death after the enrollment to evaluate clinical and laboratory data and clinical outcomes. The primary outcomes were the use of ECMO and in-hospital death after MV use, which were derived from severe ARDS. Patients with missing data, those who died from causes other than ARDS, and those who offered to withdraw from the study were excluded from the evaluation. With complete baseline clinical and laboratory data and outcome data as well as consent for participation, a total of 102 COVID-19 patients were included in the final analysis.

### Ethical considerations

This study was conducted after obtaining ethical clearance from the Institutional Ethics Board of Yokohama City University Hospital (No. B200200048).

During hospitalization, patients were provided negative and positive information regarding this study, including the purpose and contribution of this study, the use of personal information, and complications associated with blood collection, and were asked to participate in this study. Ultimately, we obtained written informed consent for participation in the study and access to medical and laboratory records from patients. Alternatively, we were unable to obtain written consent for participation from four patients who died without being weaned after being placed on a ventilator within hours of admission, which means that we used an opt-out method to obtain consent for this study from them. The study had no risk/negative consequence on those who participated in the study. Medical record numbers were used for data collection and no personal identifiers were collected or used in the research report. Data was accessed from February 9, 2020 to July 28, 2021 and access to the collected information was limited to the principal investigator and confidentiality was maintained throughout the project.

### Clinical procedure

The COVID-19 was diagnosed by reverse transcription-polymerase chain reaction (PCR) assay for SARS-CoV-2 and chest computed tomography (CT) scan. Nasal and pharyngeal swabs were also tested to exclude the influenza virus using immunochromatography assay. Routine bacterial and fungal examinations were performed. Initial laboratory investigations included a complete blood count, coagulation profile, and serum biochemical test.

### Therapeutic procedure

Respiratory function on admission was evaluated by measuring the ratio of peripheral blood oxygen saturation to fraction of inspired oxygen (SpO2/FiO2). Oxygen support (e.g., nasal cannula) was initially administered to patients with SpO2/FiO2 of less than 443. Alternatively, the need for MV or ECMO was determined according to the ratio of the partial pressure of arterial oxygen to FiO2 (PaO2/FiO2). The MV was applied if PaO2/FiO2 was less than 200 despite oxygenation. Moreover, the provision of ECMO was considered for PaO2/FiO2 less than 150 even under the use of MV.

### Data and specimen collection

We obtained clinical, laboratory, radiological, treatment, and outcomes data from electronic medical records. Two researchers also independently reviewed the data collection forms to double-check the data collected. The researchers also directly communicated with patients or their families to ascertain epidemiological and symptom data. Interview guide was not used for this purpose. Blood samples were collected within 2 hours after admission and before treatments. Laboratory test was performed immediately after collection. Moreover, for measurements of cytokines levels and SARS-CoV-2 RNA load, plasma was isolated from the same blood samples and stored at -80°C until assay. Alternatively, bronchoalveolar lavage fluid (BALF) samples were collected and stored for measurement of IL-6 level (Supplemental method in S1 File).

### IL-6 assessment

IL-6 levels were assessed using ELISA kit (R&D, D6050) according to the manufacturer's instructions.

### Quantification of viral RNA load in plasma

RNA extraction from plasma in COVID-19 patient was performed by QIAamp Viral RNA Mini Kit (QIAGEN, 52906) according to the manufacturer's instructions. The viral gene was quantified by real-time qPCR with N2 primer pairs (TaKaRa, XD0008, forward primer: `AAATTTTGGGGACCAGGAAC`, reverse primer: `TGGCAGCTGTGTAGGTCAAC`, probe: `FAM−ATGTCGCGCATTGGCATGGA-BHQ`). Viral RNA load was calculated by plotting Ct-values onto the standard curve constructed based on the standard product (NIHON GENE RESEARCH LABORATORIES, JP-NN2-PC).

### Definitions

ARDS was recognized according to the Berlin definition [15].

### Statistical analysis

Data analyses were done using the JMP ver. 12.2 software. Each value was presented as mean ± SEM, and categorical variables as frequency (%). Differences between subgroups were analyzed with Fisher's exact test (for categorical data) or the Mann-Whitney *U* test (for continuous data). Correlations between variables were analyzed by the Spearman's rank test. To evaluate the area under curves (AUCs) and cutoff values of parameters, the receiver operator characteristic (ROC) curves were constructed. The optimal cutoff was defined as value which had the best compromise between sensitivity and specificity for predicting outcome on ROC curve. To test risk stratification by combining different variables, event-free survival curves according to above or below optimal cutoffs of IL-6 and SARS-CoV-2 RNA in plasma were

constructed. Comparisons of event-free survival rates between subgroups were performed with the Log-rank test. Statistical significance was set at $P < 0.05$.

# Results

## Clinical features

Five patients without complete data and two cases of death from aspiration pneumonia or infectious aneurysm were excluded from this study. Nobody offered to withdraw from the study. Thus, the remaining 102 patients were evaluated. Table 1 shows the baseline clinical data on admission. The 70% were male, with ages ranging 30–91 years (64.9±1.5). Patients were hospitalized 5.2±0.4 days after illness onset. More than half of our study population (69%) had comorbidities including diabetes mellitus, chronic obstructive pulmonary disease, hypertension, heart diseases, and chronic kidney disease. All patients were classified on admission as moderate to severe COVID-19, as defined by the Chinese National Health Commission [16].

## Treatment and clinical outcomes

Treatments and clinical outcomes are shown in Table 1. Anti-viral agents were administered in most patients (76%). Moreover, inhaled or systemic steroids were administered in 62% or 35%, respectively of whole population. Alternatively, only 3 patients were administered tocilizumab, a recombinant humanized monoclonal antibody against the IL-6 receptor. In the context of clinical outcomes, 30 patients needed the MV 0.73±0.38 days post-admission (MV group), while the remaining 72 did not during observation. Subsequently, 5 of MV group

**Table 1. Clinical features of patients with COVID-19 in this study.**

|  | Overall (n = 102) |
| --- | --- |
| Age, year | 64.9±1.5 |
| Male, n (%) | 71 (70) |
| Onset to admission, day | 5.2±0.4 |
| Comorbidity, n (%) | 70 (69) |
| Diabetes mellitus, n (%) | 32 (31) |
| COPD, n (%) | 3 (3) |
| Hypertension, n (%) | 43 (42) |
| Heart disease, n (%) | 11 (11) |
| Chronic kidney disease, n (%) | 15 (15) |
| **Treatments** | |
| Oxygen support, n (%) | 71 (70) |
| Anti-viral agents, n (%) | 78 (76) |
| Inhaled steroid, n (%) | 63 (62) |
| Systemic steroid, n (%) | 36 (35) |
| Tocilizumab, n (%) | 3 (3) |
| **Outcomes** | |
| MV, n (%) | 30 (29) |
| ECMO use, n (%) | 5 (5) |
| In-hospital death, n (%) | 8 (8) |
| Severe ARDS, n (%) | 12 (12) |

Data are mean±SEM or number (%). COPD: chronic obstructive pulmonary disease, MV: mechanical ventilation.
ECMO: extracorporeal membrane oxygenation, ARDS: acute respiratory distress syndrome.

required ECMO for progressive refractory respiratory failure 4.8±1.1 days (range from 3 to 9 days) post-admission. One died of ARDS, while the remaining 4 recovered from ARDS and survived. Another 7 of MV group died of ARDS 13.6±1.6 days (range from 7 to 21 days) post-admission. Collectively, 12 patients reached primary outcomes composed of in-hospital death and ECMO use, derived from severe ARDS (Fatal group), while the remaining 90 did not at the 30-day (non-Fatal group).

## Early prediction for fatal outcomes

We sought biomarkers that early predict fatal outcomes after the use of MV. Initially, clinical, laboratory, and therapeutic information were compared between Fatal group and non-Fatal group (Table 2). Age tended to be higher in Fatal group compared with in non-Fatal group, without significant difference. Additionally, there was no significant difference of gender and duration from onset to admission between the 2 groups. Alternatively, more patients had diabetes mellitus in Fatal group compared with in non-Fatal group. In terms of treatment, more patients were administered inhaled steroids in Fatal group compared with in non-Fatal group. Among laboratory data on admission before treatment, serum levels of CRP, D-dimer, LDH, and AST were significantly higher in Fatal group compared with in non-Fatal group, while SpO2/FiO2 and lymphocyte counts decreased in Fatal group. There was no significant difference of GFR between the two groups. Moreover, IL-6 level and copy number of SARS-CoV-2

**Table 2. Comparisons of clinical, laboratory, and therapeutic features between non-Fatal group and Fatal group.**

|  | non-Fatal group | Fatal group | *P*-value |
|---|---|---|---|
|  | (n = 90) | (n = 12) |  |
| Age, year | 64.1±1.6 | 70.6±3.7 | 0.15 |
| Male, n (%) | 62 (69) | 9 (75) | 1.0 |
| Onset to admission, day | 5.3±0.4 | 5.2±0.8 | 0.83 |
| Comorbidity, n (%) | 59 (66) | 11 (92) | 0.0977 |
| Diabetes mellitus, n (%) | 24 (27) | 8 (67) | 0.0084 |
| COPD, n (%) | 3 (3) | 0 (0) | 1.0 |
| Hypertension, n (%) | 37 (41) | 6 (50) | 0.7569 |
| Heart disease, n (%) | 10 (11) | 1 (8.3) | 1.0 |
| Chronic kidney disease, n (%) | 12 (13) | 3 (25) | 0.3781 |
| **Treatments** |  |  |  |
| Anti-viral agents, n (%) | 67 (74) | 11 (92) | 0.2853 |
| Inhaled steroid, n (%) | 51 (57) | 12 (100) | 0.0030 |
| Systemic steroid, n (%) | 29 (32) | 7 (58) | 0.1073 |
| Tocilizumab, n (%) | 3 (3) | 0 (0) | 1.0 |
| **Laboratory data** |  |  |  |
| Lymphocyte, ×10$^6$/mL | 0.85±0.05 | 0.59±0.07 | 0.0038 |
| CRP, mg/dL | 7.6±0.8 | 17.1±3.8 | 0.0024 |
| D-dimer, μg/mL | 3.3±1.1 | 3.9±1.2 | 0.0146 |
| AST, U/L | 44.7±6.4 | 60.7±11.8 | 0.0216 |
| LDH, U/L | 331±21.8 | 473.1±58.7 | 0.0016 |
| GFR, mL/min | 65.1±3.5 | 56.3±10.2 | 0.3315 |
| SpO2/FiO2 | 359.6±13.6 | 135.9±21.6 | <0.0001 |

Data are mean±SEM or number (%). COPD: chronic obstructive pulmonary disease, CRP: C-reactive protein, AST: aspartate aminotransferase, LDH: lactate dehydrogenase, GFR: glomerular filtration rate, SpO2/FiO2: ratio of oxygen saturation to fraction of inspired oxygen.

RNA in plasma significantly increased in Fatal group compared with in non-Fatal group (Fig 1A).

Next, the ROC analyses for these significant variables showed that SpO2/FiO2, IL-6, SARS-CoV-2 RNA, CRP, and lymphocyte counts are predictive of fatal outcomes (Table 3). Particularly, SpO2/FiO2, IL-6, and SARS-CoV-2 RNA had high accuracies with AUCs > 0.80 for the risk prediction ($AUC_{SpO2/FiO2}$: 0.90, $AUC_{IL-6}$: 0.87, $AUC_{RNA}$: 0.86) (Table 3). Ultimately, combining SpO2/FiO2, IL-6 level, and SARS-CoV-2 RNA copy number showed the highest accuracy (AUC: 0.93, 95%CI: 0.83–0.98) with the best compromise between 0.92 sensitivity and 0.80 specificity (Table 3 and Fig 1B).

Clinical utility of risk stratification by combining IL-6 level and viral RNA load on admission was tested with Kaplan-Meier curves showing incidence rate of a combined event that were constructed according to above or below optimal cutoffs (Fig 1C). Even in high-risk patients with low SpO2/FiO2 (< 261), combined event rate of ECMO use and in-hospital death at the 30-day follow-up period significantly increased in patients with high IL-6 (> 49 pg/mL) and RNAaemia (> 1.5 copies/μL) compared with in those with high IL-6 or RNAaemia or without high IL-6 and RNAaemia (88% [n = 7/8] vs. 22% [n = 2/9], $P$ = 0.0097 or 8% [n = 1/12], $P$ < 0.0001, respectively). Collectively, combining IL-6 level and SARS-CoV-2 RNA copy number in peripheral blood allowed before intubation identifying COVID-19 patients at high risk for fatal outcomes after MV use, including ECMO use and in-hospital death.

## Plasma IL-6 as an indicator for immune status of diseased lung

To test whether immune status of diseased lung enable to be evaluated by peripheral information, we performed correlation analysis of IL-6 levels in plasma and BALF from critically ill patients (n = 10). Plasma IL-6 levels were positively correlated with levels in BALF (r = 0.93, $P$ = 0.001) (S1 Fig).

## Discussion

The COVID-19 pandemic rapidly increased the use of MV. Such cases often require ECMO and have a high mortality. So far, many studies have demonstrated that high levels of circulating IL-6 closely relate to negative clinical course of COVID-19 patients [11, 12]. Alternatively, SARS-CoV-2 infection of immune cells is a key trigger of cytokine release. SARS-CoV-2 RNA was detected in serum from critically ill patients at high risk for mortality who were characterized by high IL-6 [14]. High IL-6 and SARS-CoV-2 RNAaemia may reflect pathophysiologic evidence of future deterioration of COVID-19, including hyperinflammation and viral overload, thereby helping us to decide therapeutic strategy. To our knowledge, biomarkers for accurately and early identifying the risk for ECMO use, however, have yet to be established. Moreover, clinical utility of combining these biomarkers in considering ECMO use has not been elucidated. We are the first to present this data illustrating that combining IL-6 level, SARS-CoV-2 RNA load, and SpO2/FiO2 in peripheral blood on admission before intubation is likely to allow accurate risk stratification of fatal outcomes after the use of MV, including ECMO use and in-hospital death, which is of high relevance for proactive treatment and resource allocation.

Recently, multivariable mortality risk models including IL-6 and other data such as CRP, LDH, SpO2/FiO2, lymphocyte counts, and age were developed and showed higher accuracies (0.88 < AUC < 0.94) for the prediction of mortality compared with IL-6 alone [17–19]. Moreover, consistent with a recent study [14], we identified the potentiality of plasma SARS-CoV-2 RNA as an additional biomarker that predicts fatal clinical course after the use of MV for

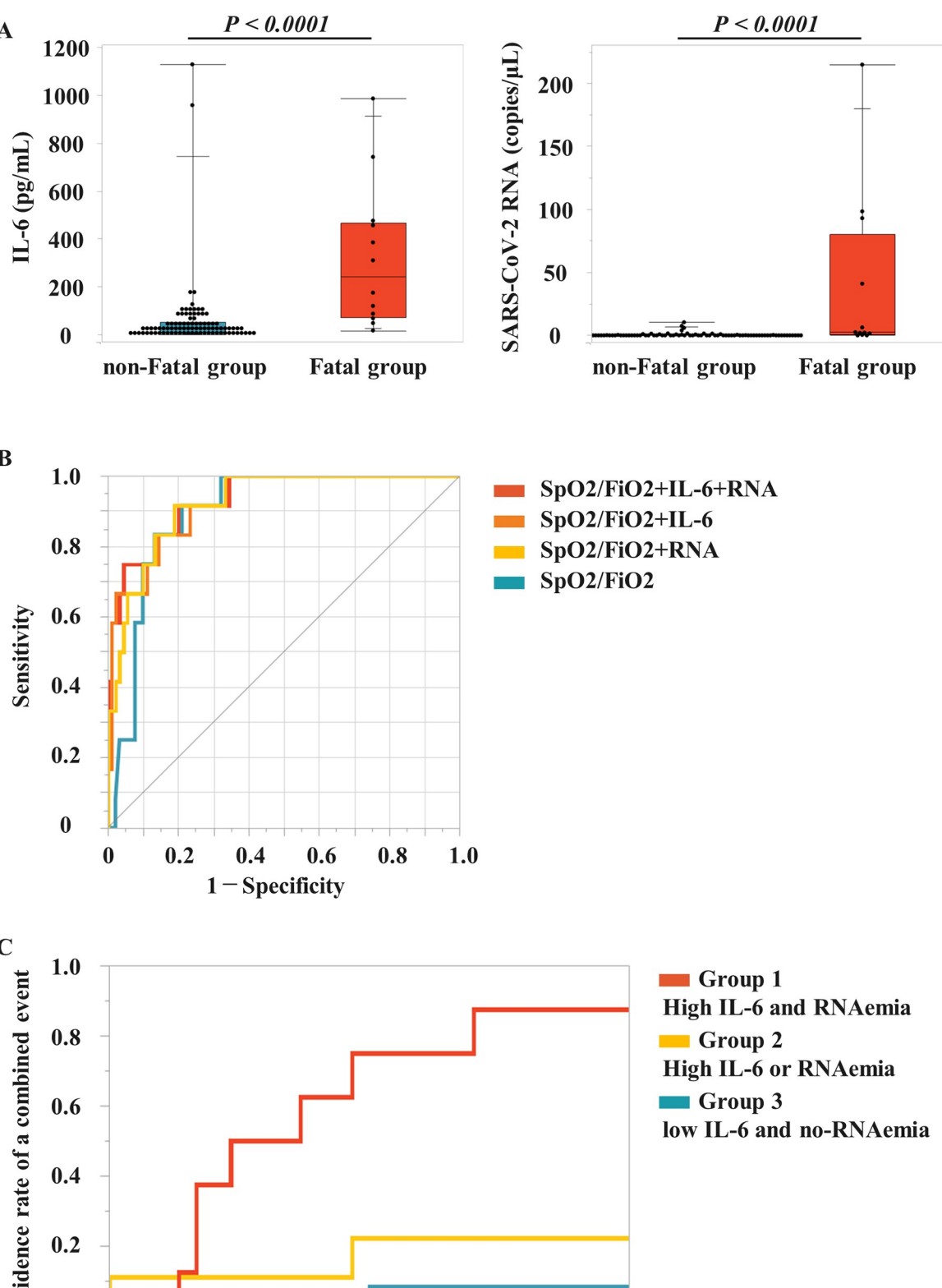

**Fig 1. Interleukin (IL)-6 levels and severe acute respiratory syndrome-coronavirus 2 (SARS-CoV-2) RNA copy number in plasma on admission, receiver operating characteristic (ROC) analysis for combined different variables, and Kaplan-Meier event-free survival analysis for a combined event of extracorporeal membrane oxygenation (ECMO) use and in-hospital death.** (**A**) IL-6 level and SARS-CoV-2 RNA copy number on admission were compared between COVID-19 patients with fatal outcomes including ECMO use and in-hospital death (Fatal group) and those without fatal outcomes (non-Fatal group). Closed circles indicate individual levels of IL-6 or SARS-CoV-2 RNA in plasma. The mean±SEM of IL-6 level and SARS-CoV-2 RNA copy number in non-Fatal and Fatal groups are as follows (IL-6: 60.9±16.3 pg/mL and 321.9±87.6 pg/mL, $P < 0.0001$, respectively; SARS-CoV-2 RNA: 0.5±0.2 copies/μL and 38.5±19.2 copies/μL, $P < 0.0001$, respectively). (**B**) ROC curves for combination of different variables on admission before intubation, including IL-6, SARS-CoV-2 RNA, and the ratio of oxygen saturation to fraction of inspired oxygen (SpO2/FiO2). (**C**) Kaplan-Meier curves showing incidence rate of a combined event of ECMO use and in-hospital death according to above or below optimal cutoffs of IL-6 level (49.0 pg/mL) and SARS-CoV-2 RNA copy number (1.5 copies/μL) in high-risk patients with low SpO2/FiO2 ($< 261$). Group 1: Both high IL-6 ($> 49.0$ pg/mL) and SARS-CoV-2 RNAaemia ($> 1.5$ copies/μL); Group 2: Either high IL-6 or SARS-CoV-2 RNAaemia; Group 3: Neither high IL-6 nor RNAemia of SARS-CoV-2.

impending respiratory failure. Interestingly, in the present study population with more elderly patients, combination of SpO2/FiO2, IL-6, and SARS-CoV-2 RNA showed a high accuracy (AUC: 0.93) for the prediction of fatal outcomes after the use of MV, even without including CRP, LDH, and lymphocyte counts. New combination of these biomarkers may help to stratify patients with critically ill COVID-19.

The combination of IL-6 and SARS-CoV-2 RNA may reflect more directly the critical pathogenesis of COVID-19 deterioration, composed of viral overload and hyperinflammation in affected lung, compared to the combination of clinical and laboratory data. The presence of nucleic acids was confirmed in the serum or plasma samples from patients with all novel coronaviruses [20]. Zheng et al. reported that, from the first week, the load of viral RNA in serum samples gradually increased, followed by a decline in the third week of the disease [21]. Lung disease derived from SARS-CoV-2 infection often emerges 7 to 10 days after symptom onset. Consistent with a recent study [14], we detected SARS-CoV-2 RNA in plasma from patients with impending respiratory failure. Viral RNAaemia can be explained by the release of viral RNA fragments from diseased lung into systemic circulation and would indicate viral overload in the lung. Alternatively, detailed single-cell RNA sequencing data from immune cells in peripheral blood as well as in BALF from patients with COVID-19 were showed [22, 23].

**Table 3. Receiver operating characteristic analyses for associations of different variables and combined variables with fatal outcomes.**

| | P-value | AUC | 95%CI | Optimal Cutoff | Sensitivity | Specificity |
|---|---|---|---|---|---|---|
| **Different variables** | | | | | | |
| SpO2/FiO2 | < 0.0001 | 0.90 | 0.81–0.95 | 261 | 91% | 79% |
| IL-6 | 0.0006 | 0.87 | 0.72–0.95 | 49 pg/mL | 92% | 73% |
| SARS-CoV-2 RNA | < 0.0001 | 0.86 | 0.68–0.95 | 1.5 copies/μL | 75% | 93% |
| LDH | 0.0643 | 0.78 | 0.66–0.87 | 281 U/L | 100% | 52% |
| CRP | 0.0012 | 0.77 | 0.63–0.87 | 5.5 mg/dL | 92% | 57% |
| D-dimer | 0.8364 | 0.72 | 0.54–0.85 | 1.1 μg/mL | 83% | 57% |
| AST | 0.4353 | 0.71 | 0.55–0.82 | 28 U/L | 100% | 43% |
| Lymphocyte | 0.0217 | 0.69 | 0.52–0.81 | $0.71 \times 10^6$/mL | 75% | 62% |
| **Combined variables** | | | | | | |
| SpO2/FiO2+IL-6+RNA | < 0.0001 | 0.934 | 0.83–0.98 | | 92% | 80% |
| SpO2/FiO2+IL-6 | < 0.0001 | 0.925 | 0.82–0.97 | | 83% | 86% |
| SpO2/FiO2+RNA | < 0.0001 | 0.924 | 0.82–0.97 | | 92% | 81% |
| IL-6+RNA | < 0.0001 | 0.912 | 0.78–0.97 | | 83% | 93% |

AUC: area under the curve, CI: confidence interval, IL: interleukin, SpO2/FiO2: ratio of oxygen saturation to fraction of inspired oxygen, SARS-CoV-2: severe acute respiratory syndrome-coronavirus 2, CRP: C-reactive protein, AST: aspartate aminotransferase, LDH: lactate dehydrogenase. P-value for fatal outcomes.

According to these reports. monocyte-derived macrophages in BALF highly expressed proinflammatory cytokines, conversely peripheral blood mononuclear cells did not express those. Thus, increasing IL-6 levels in plasma are likely to be derived from the diseased lung. Indeed, we found a close and positive correlation between IL-6 levels in plasma and its levels in BALF from patients with critically ill COVID-19. IL-6, when combined with SARS-CoV-2 RNAaemia rather than with other markers such as SpO2/FiO2, CRP, and LDH, may be a pathological biomarker indicating deterioration of primary COVID-19 rather than secondary inflammation. Multivariable mortality risk model including IL-6 and viral RNA should be developed in the prospective large cohort and extended to other viruses-induced inflammatory diseases.

If ECMO positively affects outcome of COVID-19 or not remains inconclusive. However, critically ill patients complicated by severe ARDS not having benefit from conventional treatment would be provided ECMO to overcome a life-threatening illness [24]. The ECMO use is mostly restricted to specialized centers, since careful planning, judicious resource allocation, and training of personnel to provide complex therapeutic interventions are all crucial components of an ECMO action plan. Ensuring that systems enable safe and coordinated movement of critically ill patients, staff, and equipment is essential to improve ECMO access. Low SpO2/FiO2 would be a strong predictor of ECMO use. Interestingly, Kaplan-Meier event-free survival curves that were constructed by combination of IL-6 and SARS-CoV-2 RNA showed that patients with high IL-6 and RNAaemia had an extremely high risk for ECMO use even in the high-risk group with low SpO2/FiO2 (< 261). Thus, early measuring IL-6 levels and viral RNA load in plasma together with SpO2/FiO2 may allow for preparedness of ECMO and efficient transport of high-risk patients to advanced medical center licensed to use of ECMO, contributing to optimal allocation of limited medical resources and improvement of in-hospital mortality.

Measurement of IL-6 levels and viral RNA load may also restrict the use of ECMO as well as ventilator in COVID-19 by guiding anti-viral or immunomodulatory therapies. Recent intervention studies demonstrated that dexamethasone and tocilizumab, an anti-IL-6 receptor antibody reduce MV introduction and in-hospital mortality in patients with COVID-19 [25–27]. On the other hand, the use of tocilizumab should be carefully considered, given that IL-6 is also required to prevent the replication of virus. Provided that inhibition of IL-6 receptor is too strong especially in patients with viral RNAaemia despite only slightly increase in IL-6 level, this therapy may negatively affect viral clearance and increase the risk of disease deterioration [28], resulting in an increase in MV or ECMO use. Importantly, in a recent retrospective observational study, patients with high IL-6 not treated with tocilizumab showed high mortality, as well as those with low IL-6 treated with it [29]. Thus, COVID-19 patients at high risk for ECMO use or in-hospital death, indicated by high levels of IL-6 and viral RNAaemia may be good targets for immunomodulation together with anti-viral agents. The importance of measuring IL-6 level and viral RNA load as a guide for clinical decision making should be evaluated subsequently.

The present study has several limitations. Further study is needed to validate our results, because our sample size is too small to evaluate optimal cutoffs of variables and determine predictive values of IL-6 and RNAaemia. Moreover, to define the roles of IL-6 and RNAaemia as pathological markers of COVID-19 deterioration, further investigations involving autopsy, biopsy, or BAL studies should be undertaken. To establish clinical and social importance of combining IL-6 and viral RNA as guides for treatment and patient stratification, prospective interventional study in a large cohort is needed.

In conclusion, we are the first to demonstrate that combining IL-6 and viral RNA, together with SpO2/FiO2, on admission before intubation is likely to allows early and accurately predicting fatal outcomes after the use of MV, including ECMO use and in-hospital death.

Assessing these biomarkers would help physicians formulate therapeutic strategies including anti-viral or immunomodulatory therapies and preparedness of MV or ECMO and prevent the disruption of medical care system by consolidation of limited medical resources.

## Supporting information

**S1 File. Supplemental method.**
(DOCX)

**S1 Fig. Correlation analysis between interleukin (IL)-6 levels in plasma and its levels in bronchoalveolar lavage fluid (BALF).** IL-6 levels at the same time point in plasma and BALF from patients with critically ill COVID-19 (n = 10) were measured with ELISA kit. Individual data are shown as closed circles.
(TIF)

## Acknowledgments

We thank our colleagues in the Department of Emergency Medicine, Yokohama City University and, the clinical nurses in the Advanced Care Unit, Yokohama City University Hospital for their kind assistance.

## Author Contributions

**Conceptualization:** Mototsugu Nishii.

**Data curation:** Kazuya Sakai, Yutaro Ohyama, Kento Nakajima, Taro Hiromi, Reo Matsumura, Naoya Suzuki, Hayato Taniguchi, Tsuyoshi Otsuka, Yasufumi Oi, Fumihiro Ogawa, Munehito Uchiyama, Kohei Takahashi, Masayuki Iwashita.

**Formal analysis:** Takeru Abe.

**Investigation:** Ryo Saji, Kei Miyakawa.

**Methodology:** Kei Miyakawa, Yutaro Yamaoka, Tatsuma Ban.

**Writing – original draft:** Ryo Saji, Mototsugu Nishii.

**Writing – review & editing:** Yayoi Kimura, Satoshi Fujii, Ryosuke Furuya, Tomohiko Tamura, Akihide Ryo, Ichiro Takeuchi.

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
