## [Decision Letter · Decision Letter 0]

21 May 2021

PONE-D-21-00168

Combining IL-6 and SARS-CoV-2 RNAaemia-based risk stratification for fatal outcomes of COVID-19

PLOS ONE

Dear Dr. Nishii,

Thank you for submitting your manuscript to PLOS ONE. After careful consideration, we feel that it has merit but does not fully meet PLOS ONE’s publication criteria as it currently stands. Therefore, we invite you to submit a revised version of the manuscript that addresses the points raised during the review process.

The main issues regarding the present manuscript are the small number of patients, which make results difficult to be extended to other centers, and the lack of novelty, according to existing evidence on this topic. 

We look forward to receiving your revised manuscript.

Kind regards,

Chiara Lazzeri

Academic Editor

PLOS ONE

Journal Requirements:

2)  Thank you for stating the following in the Competing Interests section:

[The authors have declared that no competing interests exist.].   

We note that one or more of the authors are employed by a commercial company: Kanto Chemical Co., Inc.

i. Please provide an amended Funding Statement declaring this commercial affiliation, as well as a statement regarding the Role of Funders in your study. If the funding organization did not play a role in the study design, data collection and analysis, decision to publish, or preparation of the manuscript and only provided financial support in the form of authors' salaries and/or research materials, please review your statements relating to the author contributions, and ensure you have specifically and accurately indicated the role(s) that these authors had in your study. You can update author roles in the Author Contributions section of the online submission form.

ii. Please also provide an updated Competing Interests Statement declaring this commercial affiliation along with any other relevant declarations relating to employment, consultancy, patents, products in development, or marketed products, etc.  

3) Thank you for stating the following in the Acknowledgments Section of your manuscript:

[Drs. Nishii and Takeuchi were recipients of the funded for this study, from the Japan Agency

for Medical Research and Development (JP19fk0108169]

 [The funders had no role in study design, data collection and analysis, decision to

publish, or preparation of the manuscript.]

4) Please provide the source and catalog numbers of the PCR assay for SARS-CoV-2 and all ELISA kits used in this study.

5)  Please provide additional details regarding participant consent. In the ethics statement in the Methods and online submission information, please ensure that you have specified:

 - whether consent was obtained

 - whether consent was informed

 - what type of consent you obtained (for instance, written or verbal, and if verbal, how it was documented and witnessed).

 - if your study included minors, state whether you obtained consent from parents or guardians.

 - if the need for consent was waived by the ethics committee, please include this information.”

6) Please provide the sequences of the N2 primers used in your study.

7) You state in your manuscript "The researchers also directly communicated with patients or their families to ascertain epidemiological and symptom data." Please state whether a interview guide was used for this purpose. If so, please provide a copy as a supplemental file.

8) In your Methods section, please provide additional information about the participant recruitment method and the demographic details of your participants. Please ensure you have provided sufficient details to replicate the analyses such as:

a) a description of any inclusion/exclusion criteria that were applied to participant recruitment,

b) a statement as to whether your sample can be considered representative of a larger population, and

c) a description of how participants were recruited.

9) PLOS requires an ORCID iD for the corresponding author in Editorial Manager on papers submitted after December 6th, 2016. Please ensure that you have an ORCID iD and that it is validated in Editorial Manager. To do this, go to ‘Update my Information’ (in the upper left-hand corner of the main menu), and click on the Fetch/Validate link next to the ORCID field. This will take you to the ORCID site and allow you to create a new iD or authenticate a pre-existing iD in Editorial Manager. Please see the following video for instructions on linking an ORCID iD to your Editorial Manager account: https://www.youtube.com/watch?v=_xcclfuvtxQ

Reviewers' comments:

Reviewer's Responses to Questions

**Comments to the Author**

1. Is the manuscript technically sound, and do the data support the conclusions?

Reviewer #1: Partly

2. Has the statistical analysis been performed appropriately and rigorously? 

Reviewer #1: Yes

3. Have the authors made all data underlying the findings in their manuscript fully available?

Reviewer #1: Yes

4. Is the manuscript presented in an intelligible fashion and written in standard English?

Reviewer #1: Yes

5. Review Comments to the Author

Reviewer #1: The authors analyzed data on a cohort of COVID-19 patients admitted in 3 hospitals in Japan in the period February-September 2020, aiming to identify risk factors for a primary composite outcome represented by ECMO and/or 30-days death. Actually, authors dedicated a large part of the paper to describe risk factors associated with MV and in-hospital death, generating a bit confusion for the reader.

Major drawbacks are:

- small sample size: cohort is relatively small (n=83) and, consequently, number of events (n=12) is too limited to support any conclusion.

- little novelty: almost all findings about conditions and biomarkers associated with negative COVID-19 outcome have been widely reported in several studies on larger populations and do not add nothing to the current knowledge. Also the increased risk of negative outcome in patients with detectable SARS-CoV-2 viremia, which remains the most interesting point of the paper, has been already reported

- authors declared that criteria for MV and ECMO were PaO2/FiO2 less than 200 and 150, respectively. Really? It seems a quite aggressive approach

6. PLOS authors have the option to publish the peer review history of their article (what does this mean?). If published, this will include your full peer review and any attached files.

Reviewer #1: No

---

## [Author Response · Author response to Decision Letter 0]

27 Jul 2021

Thank you so much for your great suggestions. We believe that your suggestions extremely improve our manuscript. According to your suggestions, we revised manuscript as attached new file in this letter.

---

## [Editor Report · Decision Letter 1]

29 Jul 2021

Combining IL-6 and SARS-CoV-2 RNAaemia-based risk stratification for fatal outcomes of COVID-19

PONE-D-21-00168R1

Dear Dr. Nishii,

We’re pleased to inform you that your manuscript has been judged scientifically suitable for publication and will be formally accepted for publication once it meets all outstanding technical requirements.

Kind regards,

Chiara Lazzeri

Academic Editor

PLOS ONE
---

## [Editor Report · Acceptance letter]

3 Aug 2021

PONE-D-21-00168R1 

Combining IL-6 and SARS-CoV-2 RNAaemia-based risk stratification for fatal outcomes of COVID-19 

Dear Dr. Nishii:

I'm pleased to inform you that your manuscript has been deemed suitable for publication in PLOS ONE. Congratulations! Your manuscript is now with our production department. 

Kind regards, 

on behalf of

Dr. Chiara Lazzeri 

Academic Editor

PLOS ONE